# Metal–Organic Frameworks as Intelligent Drug Nanocarriers for Cancer Therapy

**DOI:** 10.3390/pharmaceutics14122641

**Published:** 2022-11-29

**Authors:** Xuechao Cai, Xiaogang Bao, Yelin Wu

**Affiliations:** 1Tongji University Cancer Center, Shanghai Tenth People’s Hospital, Tongji University School of Medicine, Shanghai 200072, China; 2Department of Orthopedic Surgery, The Spine Surgical Center, Second Affiliated Hospital of Naval Medical University, Shanghai 200003, China

**Keywords:** metal–organic frameworks (MOFs), nanoMOFs, drug delivery, nanomedicine, cancer therapy

## Abstract

Metal–organic frameworks (MOFs) are crystalline porous materials with periodic network structures formed by self–assembly of metal ions and organic ligands. Attributed to their tunable composition and pore size, ultrahigh surface area (1000–7000 m^2^/g) and pore volume (1.04–4.40 cm^3^/g), easy surface modification, appropriate physiological stability, etc., MOFs have been widely used in biomedical applications in the last two decades, especially for the delivery of bioactive agents. In the initial stage, MOFs were widely used to load small molecule drugs with ultra–high doses. Whereafter, more recent work has focused on the load of biomacromolecules, such as nucleic acids and proteins. Over the past years, we have devoted extensive effort to investigate the function of MOF materials for bioactive agent delivery. MOFs can be used not only as an intelligent nanocarrier to deliver or protect bioactive agents but also as an activator for their release or activation in response to the different microenvironments. Altogether, this review details the current progress of MOF materials for bioactive agent delivery and looks into their future development.

## 1. Introduction

Drug nanocarriers are a new drug delivery technology with the development of pharmacology, biomaterial science and clinical medicine. Using nanomaterials as drug carriers can improve the absorption and utilization rate of drugs, realize efficient delivery of targeted substances, prolong the half–life of drug consumption, and reduce harmful side effects on normal tissues [1,2,3,4]. Traditional carriers such as liposomes, emulsions or micelles can carry certain drugs into tumor tissues, but their extremely low drug loading capacity (<5 wt%) limits their further clinical applications [5].

Metal–organic frameworks (MOFs), also known as porous coordination polymers (PCPs), are composed of organic ligands and metal ions or metal clusters through coordination and have an infinite network frame structure (Figure 1). In the last two decades, MOF materials have attracted considerable attention from scientists due to their versatile properties [6,7,8,9,10,11]. With their rapid development, MOFs are widely used in the fields of gas storage and separation [12,13,14,15,16,17], catalysis [18,19,20,21,22,23,24,25,26,27], chemical sensing [28,29,30,31,32,33,34], energy applications [35,36,37,38,39], biomedicine [40,41,42,43,44,45,46,47], and many other fields [48,49,50,51,52]. This inorganic–organic hybrid material not only has the characteristics of diversified structure, large specific surface area, trimmable surface, high porosity and adjustable pore size, but also can wrap target molecules such as drugs in pores as objects and release them under specific conditions, which can meet the application requirements of biomedical fields.

Compared with the traditional nanocarrier system, MOFs hold huge promise in biomedical fields due to their unique properties [53,54,55,56,57,58,59,60]. Their porous structure makes MOFs outstanding candidates for loading of various drugs; flexibility in the selection of organic ligands and metal ions enables the preparation of MOFs with inherent antitumor activity and the further design of MOF–drug synergistic systems. When the size of the MOF is reduced to the nanometer scale, i.e., nanoMOFs, they not only keep the regularity of traditional frameworks, but also have special properties of nanoparticles, such as enhanced permeability and retention (EPR) effect [61,62,63]. The past two decades have witnessed tremendous development of nanoMOF–based drug delivery systems, especially in the field of tumor therapy. Although there have been some impressive reviews on the treatment of tumors by nanoMOFs previously, a review focusing on the latest progress of anti–tumor drugs based on MOF nanocarriers is needed considering the rapid development of this field recently. This review summarizes the synthesis and latest anti–tumor application of nanoMOFs as drug carriers. It is hoped that readers will find new directions for the application and clinical translation of nanomedicine based on MOF carriers.

## 2. Synthesis of NanoMOFs

Usually, when materials reach the nanoscale, they are more suitable for biological applications, especially those that require intravenous drug treatment. Recently, there has been a lot of interest in studying the production and use of nanoMOFs, the sizes of which have at least one nanoscale dimension. As of yet, nanoMOFs have distinguished themselves from their bulk counterparts by offering distinct benefits, such as sizes that are appropriate for biomedical applications [64].

The solvothermal method is the most effective and common method for the preparation of nanoMOFs. In the initial stage, factors such as stoichiometric proportion, reaction time, pH value, temperature and the addition of surfactants are mainly considered to adjust the size of MOF crystals [65,66,67]. Currently, based on the classical theory of crystallization and LaMer’s model [68,69,70,71], and with a deeper knowledge of the nanoMOFs’ synthetic mechanism, certain techniques have been developed to control the size of nanoMOFs in accordance with the exact separation of the nucleation and growth processes. Recently, the Zhang group created a very efficient and all–encompassing method for accurately synthesizing a number of nanoMOFs by separating the nucleation and growth processes [72]. An injection pump was used to introduce the ligands and metal ions to the stirring reaction system at a predetermined dropping speed (Figure 1a). The goal of this procedure is to control the reactions’ level of supersaturation. There is a dynamic equilibrium between the consumption of the reactants and the production of the products as the reaction moves forward. According to the LaMer diagram, when reagents are continually added to the reaction system, the supersaturation degree soon rises. The termination of nucleation results from the fast fall in supersaturation degree caused by the speedy consumption of reactants that occurs during the development of nuclei. Following that, the development of MOF nanocrystals can be ensured by the ongoing addition of reactants. Consider the creation of HKUST–1 as an example. A quicker dropping speed will result in a greater quantity of nuclei being created, resulting in smaller product sizes than at a slower feed rate. When there are enough reactants, the product can develop continuously without agglomerating since the nucleation and growth processes are separated by varying the reactant concentration (Figure 1b). The ability to create more nanoMOF types, such as MIL–101(Fe), MOF–801(Zr), MIL–100(Fe), ZIF–67(Co), ZIF–8(Zn), and UiO–66(Zr), makes this synthetic approach more versatile and useful.

The microemulsion method is also an effective strategy to synthesize nanoMOFs. Microemulsions are monodisperse systems formed by thermodynamically stable, transparent water droplets in oil (W/O) or oil droplets in water (O/W) and usually consist of surfactants, cosurfactants, solvents and water. In this system, two kinds of insoluble continuous media are divided into tiny spaces by the surfactant parent molecules to form a micro–reactor, whose size can be controlled in the nanometer scale, and the reactants react in the system to form solid phase particles. In the course of mixing, monodispersed nano–droplets can develop, and the size of the droplets can be controlled by varying the quantity of surfactants. Thus, the microemulsion method can precisely control the particle size and stability of nanomaterials, the nucleation, growth, coalescence and agglomeration of nanoparticles are limited [73]. Mann and colleagues reported using this technique to create extremely monodispersed Prussian Blue nanoparticles [74], which is believed to be the earliest synthesis of nanoMOFs. This method has since been the subject of preparing nanoMOFs by the Lin group. In a micro–emulsion system made up of hexadecyl trimethyl ammonium bromide (CTAB)/water/isooctane/1–hexanol, they prepared nanosized Gd_2_(BDC)_1.5_(H_2_O)_2_ rods by mixing GdCl_3_ and bis(methylammonium)benzene–1,4–dicarboxylate [75]. Their findings show that the aspect ratios of MOF rise with the water to surfactant ratio. Additionally, the average particle size reduces when the reactant concentration rises, which may be due to an increase in the micelles that contain the reactant, which leads to an increase in nucleation sites and a decrease in particle size. This microemulsion process can also be expanded to synthesize more diverse nanoMOFs, i.e., (Gd(BTC)(H_2_O)_3_)∙H_2_O and Mn_3_(BTC)_2_(H_2_O)_6_ [76].

Recently, Cai et al. prepared a very uniform copper–based nanoMOF, i.e., HKUST–1 by constructing a microemulsion system [77]. First, the microemulsion is prepared by mixing ethanol, oleic acid, hexane and sodium hydroxide aqueous solution in specific proportions. After adding copper nitrate into the microemulsion, Cu(II) oleate is generated immediately by strong coordination between oleate ligands and bivalent copper ions. In this case, because copper ions also have dissolution properties in water, they tend to stay at the oil/water (O/W) interface. When trimesic acid ligands are added, and the temperature is elevated to a pre–set temperature, HKUST–1 nanocrystals generate gradually at the emulsion phase interface and are “protected” by oleic acid molecules with the alkyl chains on the outside, thus, the nanoparticles transfer from the O/W interface into the inner oil core, which gives the nanocrystals hydrophobic surfaces (Figure 2a). The diameter of the HKUST–1 nanoparticles increases from 30 to 140 nm when the amount of oleic acid is increased from 0.20 to 0.40 mL while maintaining the other parameters constant. Particularly, when 0.30 mL of oleic acid is added, remarkably uniform spherical HKUST–1 nanoparticles with a mean size of roughly 70 nm are created (Figure 2b–e). Additionally, this technique is used to prepare an iron(III)–based nanoMOF [78,79].

## 3. NanoMOFs as Drug Carriers

Traditional medications have weak targeting abilities and can lead to serious side effects if they build up in normal tissues [80]. Attributed to the EPR effect of nanocarriers, viable carriers are particularly important for extending the clinical use of medications. For example, two nanodrugs, doxil [81] and abraxane [82], which use liposomes and albumin as carriers, respectively, have been put on the market for their better therapeutic effects and safety. When scaled down to the nanoscale, MOFs can be exploited as promising drug carriers and have the following advantages. Firstly, by interacting with the linkers or metal clusters, drug molecules smaller than the pore width of MOFs can penetrate into the pores and be efficiently stored [80]. Secondly, through a variety of interactions, including electrostatic adsorption, coordination, and π–π stacking, cargo can be quickly loaded onto the surface of MOFs [83]. Thirdly, to prevent early drug leakage, molecules with particular groups can further build more stable covalent/dynamic interactions with functional linkers [41]. Fourthly, with increased stability and better tailorability, larger payloads including nucleic acids, peptides and proteins can be included in the framework [84]. Common MOF carriers that have been reported in the literature and patents are listed in Table 1 [85,86,87,88,89].

### 3.1. Surface Functionalization of MOFs

Nanomaterial surface engineering has long been crucial for biological applications [101]. The total effectiveness of nanoMOFs is determined by the carefully regulated modification of their exterior surfaces to meet specific needs and execute the intended function. One of the biggest functions is to increase the stability and biocompatibility of MOFs. The most prevalent and conventional technique for greatly extending the cycle duration and colloidal stability of nanoparticles is PEGylation [102,103,104].

At present, there are two commonly used post–synthesis modification (PSM) methods to modify the surface of MOFs (Figure 3). As MOFs are created via coordination bonding between organic linkers and metal ions, the first way is to modify the target molecule on the organic linker of the MOF (Figure 3a). For example, Xie et al. reported the attachment of NH_2_–poly(ethylene glycol) modified folic acid (PEGFA) with amino group in ZIF–90 via aldimine condensation. The modified nanoparticles showed enhanced biocompatibility and targeting to cancer cells [90].

The second method is to coordinate directly on the MOF surface by chelation between metal ions and target molecules (Figure 3b). Up to now, it has been reported that different functional terminal ligands coordinate with metals in MOFs, including carboxylic acid, phosphonic acid, histidine and phenyl [105,106]. The Farha team confirmed that both carboxylic acid–terminated and phosphonic acid–terminated ligands can bind to the surface of NU–1000, a Zr–based MOF [107]. Similarly, Fe, Cu–based MOFs can be easily modified by the interaction between metal and functional groups [106].

### 3.2. In Vivo Stability, Toxicity and Fate of MOFs

The stability of MOF nanoparticles under bio–related conditions strongly affects their toxicity and fate. The premise of MOF materials used in biological applications is that they should have proper physiological stability to ensure that they reach the target tissue before degradation. As MOFs are composed of inorganic and organic building blocks connected by coordination bonds, it is important to consider the possibility that the rapid degradation of MOFs in the cell chamber and the slow diffusion of degraded species may lead to a significant increase in the local concentration of metal or organic linking agents, which may lead to toxicity [108].

#### 3.2.1. In Vivo Stability

The intrinsic properties of MOFs, such as the charge density of metal ions, connection numbers of metal ions/clusters, basicity and configuration, as well as hydrophobicity of ligands, etc., have a significant impact on the chemical stability of MOFs [109]. According to the Pearson’s hard soft acid base principle, high–valent metals with a high charge density (hard acids), such as Zr^4+^, Cr^3+^, Bi^3+^, Al^3+^, and Fe^3+^, tend to combine with O donor ligands (hard bases) to produce MOFs with strong coordination bonds, which are often chemically stable. For instance, the early Cr–MIL–101 has strong acid and alkali resistance and can be stably stored for two months in aqueous solution with pH = 0–12 [110]. The highly chelating phenolates are responsible for the impressive chemical stability for the bismuth ellagate MOF, i.e., pH = 2–14, hydrothermal conditions, hot organic solvents, biological media, SO_2_, and H_2_S [108]. Even more surprisingly, the robust MIL–163, which was built from Zr ions and 1,2,3–trioxobenzene complexing group, was able to survive in PBS solution for a long time, and the thermal stability temperature reached an unprecedented 210 °C [111].

It is often said that the lack of stability hinders the potential application of MOF materials. This statement may be related to the relatively poor hydrolysis stability of some early MOFs (such as MOF–5) [112]. However, MOF materials with a certain degree of instability have a precise advantage in biological applications. For example, the studies on the stability of MOF materials such as MIL–100 under simulated physiological conditions showed that these MOFs did not dissolve quickly but could last for a long time under physiological conditions [113]. In addition, some in vivo experiments showed that MOFs of ferric carboxylate are biodegradable; irons can be recycled, and the linkers are relatively easy to remove.

#### 3.2.2. The Toxicity of MOFs

There are many factors affecting the toxicity of MOF materials, mainly depending on their physicochemical properties, including MOF stability, chemical composition, surface functionalization, size, etc. A classic piece of work was performed by Horcajada’s research group, which prepared 14 different MOF materials and tested their cytotoxicity [91]. Two cell lines (J774 and HeLa) were used to evaluate the cytotoxicity of these nanoMOFs using the MTT method, which showed that their low toxicity values were similar to those of other currently commercialized nanosystems. It is observed that the cytotoxicity depends largely on the components of MOF, for instance: (i) the properties of metals, the toxicity of iron–based MOFs is less than that of Zr– or Zn–MOF nanoparticles, and (ii) the constitutive organic linker, and the hydrophobic–hydrophilic balance is an important parameter.

#### 3.2.3. The Fate of MOFs

Cai et al. studied in detail the metabolism of Cu–Tz–1 MOF in vivo [114]. In the first 12 h after intravenous tail injection, the MOF nanoparticles were mainly accumulated in liver and spleen (Figure 4a). After 24 h, the amounts of nanoparticles began to decrease in the tested organs. Under the action of bile, nanoMOFs accumulated in the liver were metabolized out of the body. After seven days, the degraded nanoparticles could be excreted through kidney to form urine. At 14th day, the nanoparticles reached a maximum with the excretion of urine. Thereafter, the discharge amount was gradually reduced, and after 30 days, the nanoparticles were discharged at a high total rate of ≈90% through feces and urine (Figure 4b). This work demonstrates that certain MOF materials are biodegradable and can be excreted out of the body.

### 3.3. NanoMOFs for Small Molecule Delivery

As a carrier of small molecule drugs, one of the advantages of MOF materials is the high loading capacity [92]. In 2006, Horcajada et al. constructed two kinds of cubic zeolite MOFs, i.e., MIL–100 and MIL–101 [93]. These materials showed a drug loading capacity of up to 60% for ibuprofen, which was obviously superior to the loading capacity of traditional carriers [115]. It proved the high potential of porous MOFs in drug loading for the first time.

Because of the high biological toxicity of chromium ions, the same team then synthesized a series of low–toxicity porous ferric carboxylate MOFs, and these materials can be loaded with up to nine common anticancer drugs (Figure 5) [5]. Among these MOFs, the loading of the chemotherapy drug busulfan in the mesoporous Fe–MIL–100 was as high as 25 wt%, which was five times and 60 times higher than that of polymer nanoparticle system and liposome, respectively. The loading of azidothymidine triphosphate and cidofovir on MIL–101–NH_2_ reached an unprecedented 42 wt%. More satisfactorily, after modification with polyethylene glycol (PEG), the physiological stability of MOF materials is enhanced, so the MOF nanocarriers can realize controlled release without “burst effect”. The continuous release time of DOX in Fe–MIL–100 can reach 13 days, and the release rate can reach 100% (Figure 5b), which far exceeds that of other drug carriers in the past. Since then, the research on MOFs as drug carriers has developed rapidly.

The controlled release of loaded drugs has always been the focus of researchers and MOF materials show unparalleled talent in this field. Sun et al. discovered the pH–responsive drug release performance of zeolite imidazole framework–8 (ZIF–8) for the first time [116]. The pH–responsive drug release experiment with ZIF–8 loaded with 5–Fu showed that 5–Fu was not released or slightly released in PBS (pH = 7.4), and the release rate in acetate buffer (pH = 5.0) was significantly increased, and more than 45% of 5–Fu was released within about 1 h. Afterwards, Zheng et al. reported a method that encapsulated anticancer drug doxorubicin (DOX) during the synthesis of ZIF–8 nanocrystals by a one–pot process (Figure 6a) [117]. Due to the strong interaction between drug molecules and zinc ions, the loading amount of the DOX molecules can be adjusted, and they are uniformly dispersed within the ZIF–8 nanoparticles. Confocal microscope was employed to test the uptake of free DOX and DOX@ZIF–8 in MDA–MB–468 cells (Figure 6b). The data indicated that free DOX access the nucleus within two hours and accumulated in the nucleus, as a contrast, DOX@ZIF–8 nanoparticles arrived in cytoplasm first. After 24 h, most cells died under the treatment of DOX@ZIF–8, and only cell fragments were observed. Compared to DOX alone, DOX@ ZIF–8 showed pH–responsive drug release, and its therapeutic effect was enhanced.

### 3.4. MOFs for Gas Molecule Delivery

Some endogenous therapeutic gases having diverse biological effects, such as carbon monoxide (CO), nitric oxide (NO) and oxygen (O_2_), hold considerable promise for treating a variety of medical issues. Because they can be removed before reaching the intended target location, these therapeutic gases’ short half–life in human tissues severely limits their therapeutic usefulness [118]. They must perform the necessary activities in a controlled manner since their biological functions are also significantly reliant on concentration and location. Loading these gas drugs into MOFs is an effective way to solve these problems.

#### 3.4.1. MOFs for CO Delivery

The concentration of CO has a significant impact on its biological function. In order to obtain the intended therapeutic effect, CO must be supplied in a safe and manageable manner [119]. Yao et al. loaded CO donor (MnBr(CO)_5_) and anti–cancer drug DOX on a multifunctional nano–platform composed of a magnetic carbon (MC) core and Fe–MOF shell [120]. Fe–MOF acts as a drug carrier. MC has photothermal effect, which can convert near infrared (NIR) light into heat and induce the release of loaded drugs (Figure 7a). Finally, compared with the control group and other treatment groups, the combined release of CO and DOX stimulated by NIR light showed the best anti–tumor effect (Figure 7b).

#### 3.4.2. MOFs for NO Delivery

Morris and his collaborators performed some pioneering work in storing and delivering NO with MOFs. They showed that (M_2_(C_8_H_2_O_6_)(H_2_O)_2_)·8H_2_O is an ideal carrier for NO storage and transportation because of its high NO adsorption capacity (about 7 mmol of NO/g of MOF) and good storage stability [121]. More excitingly, the weight adsorption measurements of HKUST–1 on NO at 196 K (1 bar) revealed that it had an adsorption capacity of up to roughly 9 mmol of NO/g of MOF, which is much greater than that of other porous solids [94].

For biological applications of NO, loading its precursors into MOFs is an effective strategy. Zhang et al. used Mn–porphyrin MOF as a carrier to load S–Nitrosothiol (SNO) for heat–sensitive NO generation [122]. The nanocomposites were first gathered at the site of mouse tumors after intravenous injection. Then, the NIR laser produced simultaneous controlled NO release and PTT for effective one–step synergistic treatment. It is worth pointing out that the MOF–SNO composites showed substantial tumor suppression effectiveness when compared to treatment with DOX for a simple drug–resistance model, demonstrating the great advantage of MOF as a drug carrier.

#### 3.4.3. MOFs for O_2_ Delivery

Tumor hypoxia is widespread [123] and it is necessary to deliver O_2_ to the tumor site to relieve hypoxia. Some MOF materials can adsorb O_2_ and transport O_2_ into tumor tissues to overcome hypoxia, thereby enhancing the therapeutic effect on tumors [114]. For example, Xie et al. reported an O_2_–loaded multifunctional nanoplatform, namely O_2_–Cu/ZIF–8@Ce6/ZIF–8@F127 (OCZCF), which can overcome hypoxic tumor microenvironment and increase the generation of ROS under irradiation for enhanced tumor therapy (Figure 8a) [124]. Specifically, Cu–doped ZIF–8 greatly enhanced the oxygen adsorption performance. Compared with the pristine ZIF–8, the oxygen adsorption capacity of Cu–doped ZIF–8 is doubled and the oxygen can be released quickly in acidic environments (Figure 8b,c). In addition, the strong chemisorption of photosensitizer Ce6 with MOF matrix ensures its successful coating and sufficiently high loading rate. The large amount of released oxygen alleviated hypoxia, which greatly promoted the ability of photosensitizer to produce singlet oxygen. Taken together, this nanoplatform showed a good tumor–killing effect both in vitro and in vivo (Figure 8d,e).

### 3.5. NanoMOFs for Photosensitizer Delivery

Photodynamic therapy (PDT) is a kind of therapeutic method that uses the photochemical reaction of photosensitizer in organisms after receiving specific laser irradiation, which can achieve the effect of killing tumor or removing infected microorganism [125,126,127,128]. During this process, photosensitizer plays a crucial role. However, photosensitizer has poor water solubility and low utilization rate in organisms, which limits its development. Similar to the situation of the small molecule drugs, MOF materials are excellent carriers of photosensitizers for tumor treatment [95,129,130,131]. Theoretically, the metal centers and ligands used to construct MOF materials are infinite, among which includes the use of porphyrin molecules or their derivatives as ligands [46]. In 2014, Lin group first used hafnium–porphyrin nanoMOF as a photosensitizer for photodynamic therapy. DPB-UiO was formed by HfCl_4_ and a new kind of porphyrin derivative, 5,15-di(p–benzoato)porphyrin (H_2_DBP) [132]. After injecting the nanoMOFs into the tumor, a large amount of singlet oxygen was produced in the tumor under the irradiation of 660 nm light. After treatment, the tumor volume was reduced by nearly 98% compared with the original volume, while the single injection of H_2_ DBP photosensitizer had no effect. Compared with the traditional organic nanoparticle carrier, nanoMOF with porphyrin as an organic ligand can reduce the self–quenching effect caused by light and facilitate the diffusion of reactive oxygen species, so it can multiply the efficiency of photodynamic therapy.

Since the size of nanoparticles affects their circulation, tissue distribution and cellular uptake [133], the construction of size–controlled synthesis methods has attracted much attention. In view of this, Zhou group successfully synthesized different sizes of zirconium–porphyrin nanoMOF, i.e., PCN–224, with multi–particle size distribution of 30–190 nm, and proposed that if the formation process of nanoMOF is regarded as the replacement reaction of metal cluster ligand, the concentration of PCN–224 monomer can be controlled by adjusting the concentration of each component in the system (Figure 9a,b) [96]. The particle size of the prepared nanoMOF is smaller when the monomer concentration is higher in the nucleation process. The experimental results showed that compared with the particles of 30, 60, 140 and 190 nm, the particles of 90 nm were easier to be taken up by the cells, and the porphyrin derivative TCPP could be efficiently retained in the cytoplasm. Using in vitro assays, the killing rate of HeLa cells was up to 80% after 30 min irradiation with a 420 or 630 nm laser (Figure 9c). In addition, by further modifying the folate targeting group on the surface of PCN–224, the photodynamic therapy efficiency of PCN–224 could exceed 90% (Figure 9d), which is similar to the effect of the porphyrin series photosensitizer used in clinical practice.

### 3.6. NanoMOFs for Nucleic Acid Delivery

After 20 years of intensive research, gene therapy is now one of the most promising ways to treat cancer. However, naked RNA/DNA molecules are too large to pass through cell membranes and are often rapidly degraded by serum nucleases in the blood [134], the lack of an ideal drug delivery system limits the potential of gene therapy. In recent years, small interfering RNA (siRNA) has been used to alter gene expression in cancer cells for efficient cancer therapy [135]. NanoMOFs have been used as an effective nanocarrier to load siRNA, and they can protect siRNA from being cleared or degraded before it acts in target cells. Lin’s research team reported that MOF nanocarrier co–delivered cisplatin and pooled siRNA to enhance the chemotherapy effect of drug–resistant ovarian cancer cells [136]. siRNA and cisplatin prodrugs were loaded on the surface and internal pore channels of a UiO–type Zr–MOF nanosheets, respectively. The data showed that nanoMOF protected siRNA from degradation and increased cellular uptake of siRNA to silence multidrug resistant genes, thus, greatly promoting the chemotherapy effect of cisplatin. Similarly, the Morris group [137] created a nucleic acid–MOF nanoparticle conjugate, based on azide–functionalized UiO–66–N_3_, which provides a prospect for gene therapy. The siRNA was applied to the surface of the hexagonal plate by coordination with metal sites, which protected the siRNA from nuclease degradation, increased its cellular uptake, and promoted the escape of siRNA from endosomes, silencing multidrug resistance genes. In addition, biomimetic mineralization and co–precipitation methods are used to encapsulate plasmid DNA macromolecules into ZIF–8 and ZIF–8 polymer vectors. Both systems have good plasmid DNA loading, release and protection ability, and show good performance in intracellular gene delivery and expression [138].

Very recently, our group constructed a novel PolyIC@ZIF–8 nanoplatform [139], which combines the structural and functional characteristics and advantages of oncolytic viruses, endowing the nanoparticles with oncolytic virus–like functions (Figure 10). It is worth mentioning that ZIF–8 promotes the recruitment and activation of T cells in a tumor antigen–independent manner in response to the release of Zn^2+^ from the tumor microenvironment, thus achieving highly effective tumor immunotherapy. This study demonstrated that ZIF–8, with virus–like coat structure, can protect double–stranded RNA (PolyIC) efficiently into tumor cells, up–regulate MDA–5 to induce tumor cell apoptosis, and then induce DC maturation, promote tumor antigen presentation to T cells, and promote T cell recruitment and activation in a classical antigen–dependent manner. More importantly, Zn^2+^ released by ZIF–8 can not only directly promote the expression of CXCL9/10/11 in DC cells, but can also enhance the enrichment of T cells into tumor areas. It can also directly induce the phosphorylation of ZAP–70 to activate T cells and promote T cells to produce more IFN–γ to kill tumors. In this innovative strategy, the process of Zn^2+^–mediated T cell recruitment and activation is independent of tumor antigen and combines with PolyIC antigen–dependent pathways to jointly promote tumor cell killing and achieve efficient tumor therapy. The new strategy of ion–enhanced oncolytic virus–mediated tumor immunotherapy proposed in this research provides a new perspective for the application of metal ions in tumor therapy, further expands the application scope of “ion interference therapy” [97,140], and is expected to provide a new idea for ion regulation of tumor microenvironment, so as to realize high–efficiency tumor treatment.

### 3.7. NanoMOFs for Enzyme/Protein Delivery

Protein is composed of amino acids in the way of “dehydration condensation” of polypeptide chain after folding to form a certain spatial structure of the material. They perform a wide range of tasks, including DNA replication, metabolic process catalysis, and molecular transport. Proteins have a tough time naturally crossing membranes without losing structural integrity due to their huge size, charged surfaces, and environmental sensitivity. MOF nanoparticles for intracellular protein delivery have gained more attention recently as a means of using proteins therapeutically [98,141].

Recently, Mao group reported an ATP–responsive zeolite imidazole framework–90 (ZIF–90) as a nanocarrier for cytoplasmic protein delivery and CRISPR/Cas9 genome editing (Figure 11) [142]. Imidazole–2–carboxyaldehyde and Zn^2+^ self–assembled with protein to form ZIF–90/protein nanoparticles, which effectively coated protein. The results showed that in the presence of ATP, ZIF–90/protein nanoparticles degraded and released protein because of the competitive synergy between ATP and Zn^2+^ of ZIF–90. Further studies showed that ZIF–90/protein nanoparticles can deliver all kinds of protein into cytoplasm regardless of the size and molecular weight of protein. The delivery of toxic RNase A can effectively inhibit the growth of tumor cells, while the delivery of genome editing protein Cas9 can effectively inhibit the expression of green fluorescent protein (GFP) in HeLa cells, with an efficiency of 35%. In view of the fact that ATP is up–regulated in diseased cells, it is expected that the delivery of ATP–responsive proteins can discover new opportunities for protein delivery and targeted disease treatment of CRISPR/Cas9 genome editing.

Catalase DNAzymes (CAT Dz) are known to be effective therapeutic agents for gene therapy, but they are not currently used in biologically useful ways due to their ineffective intracellular delivery and insufficient cofactor supply. Recently, our group synthesized ZIF–82–based nanoparticles loaded with CAT Dz on FeCysPW surface (Figure 12) [143]. Among them, the shell ZIFs can not only effectively protect and deliver DNAzymes to tumor cells, but also provide them with “cofactors” for effective gene silencing. After the nanoplatform is ingested by tumor cells, ZIF–82@CAT Dz can release Zn^2+^, imidazole ligands with electron affinity (i.e., 2–nitroimidazole and 1H– imidazole–4–nitrile), and CAT Dz in response to acid tumor microenvironment. The released Zn^2+^ can help CAT Dz silence catalase and promote the accumulation of H_2_O_2_ in tumor cells. At the same time, the dissociated electrophilic ligand can rapidly consume glutathione in hypoxic tumor cells, resulting in the imbalance of redox homeostasis and the accumulation of H_2_O_2_ in hypoxic tumor cells. The inner core FeCysPW efficiently converts endogenous H_2_O_2_ into hydroxyl radicals with stronger toxicity by Fenton reaction, thus significantly enhancing the chemodynamic therapy (CDT) efficacy of hypoxic tumor and realizing the efficient treatment of hypoxic tumor (Figure 12b,c).

### 3.8. NanoMOFs for Combined Synergistic Treatment

In recent years, there are more and more research reports on combination cancer therapy based on nanoMOF carriers. This kind of research has become a new trend in clinical oncology because of its synergistic and efficient therapeutic effects and reduction of side effects caused by different therapeutic methods [99,144,145,146]. In addition to using MOF materials as drug carriers, using the tunability and structural regularity of MOF synthesis, designing various functional ligands, nanoparticles and biomolecules for post–synthesis modification can regulate the heterogeneous structure of MOFs and integrate different biomolecules into a single frame in layers, thus realizing multifunction [147]. Our group has also performed some related work. For example, we recently reported the load of plasma amine oxidase (PAO) into Fe–MIL–100 nanoparticles with high Fenton reaction activity, and then modified the nanoparticles with polyvinylpyrrolidone (PVP) to construct a new polyamine–activated Fe–MIL–100@PAO@PVP nanoplatform (Figure 13) [148]. After the nanoparticles reach the tumor site, the excess polyamine molecules are rapidly decomposed by enzyme–catalyzed reaction, and a large amount of H_2_O_2_ and highly toxic acrolein are generated. H_2_O_2_, as the substrate of CDT [100,149,150,151,152], significantly increased the yield of strong oxidative ∙OH in cells, and caused the oxidative damage of tumor cells. At the same time, acrolein can activate carbonyl stress reaction in tumor cells and inhibit the expression of GPX4 and DNA repair protein, leading to severe lipid peroxidation and DNA damage, and finally cause the death of tumor cells. Employing a nanoMOF as the carrier, the excess polyamine in tumor was used for the first time in this study, and a new strategy of polyamine–activated carbonyl stress was put forward, which realized the efficient treatment of tumor oxidative damage. Polyamine–activated carbonyl stress, a new tumor treatment strategy, not only effectively overcomes the defect of low efficiency of oxidative damage caused by oxidative resistance of tumor cells, but also provides a new reference method and research ideas for the design of other new anti–tumor agents.

In addition, we recently constructed a MOF nanoplatform to limit cancer cell metastasis. Briefly, a new type of hydrogen ion nano–donor, UCNP@MIL–88B@PA (UMP for short) was prepared [153], in which upconversion luminescent nanoparticles (UCNPs) are the functional core, Fe–MIL–88B is the outer shell, and photoacid molecules (PA) are loaded in the pores of Fe–MIL–88B (Figure 14a,b). Under the irradiation of a 980 nm laser, the loaded PA molecules are excited by up–converted photoexcitation to release hydrogen ions and instantly increase the hydrogen ion concentration in tumor cells. More importantly, hydrogen ions can bind to actin cleavage protein in tumor cells, which significantly inhibits the activity of F–actin, thus regulating the formation and motor function of pseudopodia in tumor cells, reducing the number of axopodia, leading to the disintegration and collapse of actin cytoskeleton, and finally significantly reducing the movement, invasion and migration ability of U87–MG tumor cells (Figure 14c–e). It is worth mentioning that with the increase of hydrogen ion concentration in tumor cells, the iron–based Fenton reaction ability of Fe–MIL–88B can be significantly improved, the production of hydroxyl radicals can be increased, and acid–enhanced CDT can be realized. Finally, UMP with near infrared light showed a good anti–tumor therapeutic effect (Figure 14f,g). From the unique point of view of interfering with the biological function of tumor cells’ pseudopodia, this research work developed a new strategy of anti–tumor metastasis therapy based on the regulation of tumor cells’ motility acid, which not only opened up a new perspective of hydrogen ion regulation for high–efficiency tumor therapy, but also opened up a new idea for nano–functional materials to be used in the interdisciplinary field of life medicine.

## 4. Conclusions and Perspectives

NanoMOFs are emerging porous inorganic–organic hybrid crystal materials and have important scientific research value in the field of drug delivery. Compared with traditional nano–drug carriers, the unique framework structure and composition are beneficial to their applications. This review briefly describes the synthesis and development of nanoMOF materials as drug carriers to load chemotherapy drugs, gaseous molecules, photosensitizers, nucleic acids and proteins. NanoMOFs can bind drugs in various ways, and the abundant particle structure and drug loading mode provide the possibility of combination of various treatment methods. Although the development of nanoMOFs in the field of biomedicine is still in the initial stage, it can be foreseen that nanoMOFs integrating cell targeting, drug delivery, molecular imaging and tumor therapy can emerge.

For the perspectives of the field, the biocompatibility of nanoMOFs is one of the important factors restricting their development. In fact, the biological applications of MOF materials are still in preclinical research, but more and more studies are proving their clinical potential. NanoMOFs with suitable physiologic stability can minimize the cytotoxicity and improve their bioavailability. Because the toxicity of MOF materials depends largely on their composition, in order to achieve this goal, the first choice is to use non–toxic metal ions and ligands. Bio–MOFs may be a good candidate for potential clinical applications [154,155]. For example, the MOF SU–101, which is composed of bismuth ions and ellagic acid ligands, demonstrated excellent biosecurity [108]. In addition, biological endogenous ligands are suggested. For example, amino acid–MOFs [156,157] with proper stability have great clinical potential in drug delivery. Finally, the kinetics of drug loading and release, in vivo toxicity, the mechanism of drug degradation, and the pharmacokinetics of nanoMOFs all require more studies, as well as logically creating MOF–drug conjugates that have better biostability, biocompatibility, and therapeutic effectiveness. In conclusion, MOFs have special qualities and hold considerable potential for the intracellular delivery of drugs. To fully exploit the promise of MOFs as drug delivery systems in clinical applications, efforts should be concentrated on resolving the aforementioned difficulties in the future.

## Data Availability

Not applicable.

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
