# Peer review of "Metal–Organic Frameworks as Intelligent Drug Nanocarriers for Cancer Therapy"

_pharmaceutics, 2022, doi:10.3390/pharmaceutics14122641_

Round 1

Reviewer 1 Report

The review is interesting, nonetheless, it comes short in certain aspects.

1.      The review seems to be centered in the authors’ work and the work by the Lin group. Please survey the literature to include other works.

2.      A general structure of the MOFs should be presented at the beginning of the manuscript.

3.      A critique should be included with respect to the use of toxic molecules to synthetize MOFs, such as CTAB, hexane, and ligands that could end up removing ions from enzymes or from the circulation. The use of toxic and expensive ions should also be criticized.

4.      A section dealing with functionalization (dealing with targeting and passivation with PEG for example) of MOFs must be included

5.      A survey in the literature must be run to include a section of toxicity, stability, and fate of MOFs

6.      The section of gas delivery using MOFs is short, include other works.

7.      Modify conclusions accordingly, once the new sections are included in the manuscript

Other general comments:

Page 1, line 14, do not use colloquial language (and so on)

Page 1, line 28-30, poor clarity, please rewrite (Drugs enter the human body through the carrier, the carrier adsorbed, wrapped and bonded to the drug, making the drug release site, speed and mode selective and controllable, realizing the sustained release and targeted delivery of drugs, so as to give better play to the efficiency of drug treatment)

Page 1, line 34, change “bypass, etc., which limits” to “bypass, etc., limit”

Page 1, line 32-35, the statement “Traditional carriers such as liposomes, emulsions or micelles can carry certain drugs into tumor tissues, but their serious low drug loading (< 5 wt%), unsatisfactory bioavailability, rupture release, poor biobarrier bypass, etc., which limits their further clinical applications” has some misinformation. Search for carries approved by the FDA, that will make them “traditional”, liposomes are certainly approved especially after COVID. Liposomes have no poor biobarrier bypass, I do not know about the others (emulsion and micelles). Provide a correct statement then.

Page 2, line 69, change “is previously the most” to “is the most”

Page 2, line 76, change “Recently, Zhang group” to “Recently, the Zhang group”

Page 2, line 88, change “than a slower” to “than at a slower”

Page 4, line 113, change “by Lin group” to “by the Lin group”

Page 4, line 122, change “Recently, Lin group” to “Recently, the Lin group”

Page 5, line 147, the statement “Traditional medications have weak targeting abilities and can lead to serious side effects if they build up in normal tissues” is also true for untargeted MOFs. Targeting is required for the MOFs to reach a specific release site. None of this is mentioned in the paragraph.

Page 6, line 165-167, add references to support this statement: “These materials showed a drug loading capacity of up to 60% for ibuprofen, which was obviously superior to the loading capacity of traditional carriers”

Page 6, line 176, please explain how the attachment of PEG reduces the burst effect

Page 9, line 264, change “In vitro assays” to “Using in vitro assays”

Reviewer 2 Report

This manuscript summarises recent studies using MOFs as medicine carriers for controlling medicine delivery. To my knowledge, most of the relevant studies still remain in research labs, and are not commercialised. Can authors discuss the criteria required for clinic drug delivery particularly if MOFs are adopted? What are the advantages of MOFs when it is used for drug delivery and how to avoid their weakness? Generally, when talking about nano MOF, I assume it refers to the nano size. Although lots of publications use the same definition, the Nano MOF is quite ambiguous. the size can distribute in a wide range from a few nanometers to hundreds of nanometers or close to micrometres. At low bound, the MOF can be reduced to a molecular cage compound, does that make MOF more useful as a medicine carrier? For gas drug delivery, the pioneering work was done by the Morris group (JACS 2008, 130(31),10440). "only when materials reach the nanoscale can they be better used for biological applications", I feel, this description is not accurate. The use of materials completely depends on the conditions if they are for clinical applications. Drug delivery can be in the body, and can also be applied to topical therapies. Accordingly, the requirements for delivering materials are different. Can authors discuss this?

Reviewer 3 Report

This manuscript is well written and timely.

Author Response

Thank you very much.

Reviewer 4 Report

The present review manuscript entitled "Metal-Organic Frameworks as Intelligent Drug Nanocarriers for Cancer Therapy" involved the bibliographical analyses from the MOFs synthesis to their uses as drug carriers focus on potential cancer therapies.

The introduction is according to the aim of this research, and it is appropriate for the desire analyses/conclusions the authors reported.

Although, there are current reported analyses that the authors didn’t consider in order to complete the present review such as:

1)Materials 2021 Dec; 14(23): 7277. Doi:10.3390/ma14237277. Metal–Organic Frameworks (MOFs) for Cancer Therapy

2)International Journal of Nanomedicine. Recent Progress of Metal-Organic Framework-Based Photodynamic Therapy for Cancer Treatment. DOI https://doi.org/10.2147/IJN.S362759

3)Application of organic frame materials in cancer therapy through regulation of tumor microenvironment. https://doi.org/10.1016/j.smaim.2022.01.006. Smart Materials in Medicine. Volume 3, 2022, Pages 230-242

4)Metal-organic frameworks for hepatocellular carcinoma therapy and mechanism. Frontiers in Pharmacology https://doi.org/10.3389/fphar.2022.1025780

5)Porphyrin–palladium hydride MOF nanoparticles for tumor-targeting photoacoustic imaging-guided hydrogenothermal cancer therapy. Nanoscale Horiz., 2019,4, 1185-1193. https://doi.org/10.1039/C9NH00021F

6)Cancer cell membrane-camouflaged MOF nanoparticles for a potent dihydroartemisinin-based hepatocellular carcinoma therapy. https://doi.org/10.1039/C9RA09233A. RSC Adv., 2020,10, 7194-7205

7)Zhao, D., Zhang, W., Yu, S. et al. Application of MOF-based nanotherapeutics in light-mediated cancer diagnosis and therapy. J Nanobiotechnol 20, 421 (2022). https://doi.org/10.1186/s12951-022-01631-2

8)Metal-Organic Framework Nanoparticle-Based Biomineralization: A New Strategy toward Cancer Treatment. Theranostics 2019; 9(11):3134-3149. doi:10.7150/thno.33539

I suggest including the missing information due to the current state of the art and the increasing number of reviews in this interesting field of research.

Moreover, the information they described is supported with clear and logical images/figures/tables that summarize all the required data for the discussion item previously reported in the bibliographical references. According to this point, it should be necessary to add some comparative table analysis of the present MOFs developments for cancer therapy to summarize the information (advantages, disadvantages, biocompatibility, synthesis methods, etc.), please include current patents. This point should be necessary in order to produce a novel review in this area.

Besides, I encourage the authors to check some mistakes such as (highlighted in yellow, pdf attached):

- Please remember to use italics for in vivo, in vitro.

- Please homologate the size and the style of the graphics throughout the manuscript.

Furthermore, it would be interesting and reasonable the authors could indicate a potential improvement to obtain better biocompatibility properties to achieve this important goal (Conclusions and Perspectives, highlighted in green, pdf attached).

Finally, I would like to invite the authors to include the abbreviation list of words at the end of this manuscript.

Round 2

Reviewer 1 Report

Most of my concerns were addressed by the authors. Thank you for that. I have a couple of comments though. 

In line 199 of the revised manuscript you added a section dealing with stability. I did not mean this kind of stability. I meant nanoparticle stability. The stability of MOFs in terms of maintaining their presence as single entities upon conducting an application. For example, can nanoMOFs exist in the presence of PBS buffer? Please modify section 3.2.1 accordingly.

In line 169, section 3.1 was added, which I think is fine, a Figure depicting surface modification of MOFs will help a lot, though. Please add this figure to help understanding this section.   

Reviewer 4 Report

The authors performed all the suggested corrections, thanks.

Author Response

Thank you very much for your affirmation of our manuscript.